# Challenges in the Vaccination of the Elderly and Strategies for Improvement

Gatot Soegiarto [1,2,*] and Dewajani Purnomosari [3]

1    Allergy and Clinical Immunology Division, Department of Internal Medicine, Dr. Soetomo Academic General Hospital, Faculty of Medicine, Universitas Airlangga, Surabaya 60286, Indonesia
2    Master Program in Immunology, Postgraduate School, Universitas Airlangga, Surabaya 60286, Indonesia
3    Department of Histology and Cell Biology, Faculty of Medicine, Public Health and Nursing, Universitas Gajah Mada, Yogyakarta 55281, Indonesia; d.purnomosari@ugm.ac.id
*    Correspondence: gatot_soegiarto@fk.unair.ac.id

**Abstract:** In recent years, the elderly has become a rapidly growing proportion of the world's population as life expectancy is extending. Immunosenescence and inflammaging contribute to the increased risk of chronic non-communicable and acute infectious diseases. Frailty is highly prevalent in the elderly and is associated with an impaired immune response, a higher propensity to infection, and a lower response to vaccines. Additionally, the presence of uncontrolled comorbid diseases in the elderly also contributes to sarcopenia and frailty. Vaccine-preventable diseases that threaten the elderly include influenza, pneumococcal infection, herpes zoster, and COVID-19, which contribute to significant disability-adjusted life years lost. Previous studies had shown that conventional vaccines only yielded suboptimal protection that wanes rapidly in a shorter time. This article reviews published papers on several vaccination strategies that were developed for the elderly to solve these problems: more immunogenic vaccine formulations using larger doses of antigen, stronger vaccine adjuvants, recombinant subunit or protein conjugated vaccines, newly developed mRNA vaccines, giving booster shots, and exploring alternative routes of administration. Included also are several publications on senolytic medications under investigation to boost the immune system and vaccine response in the elderly. With all those in regard, the currently recommended vaccines for the elderly are presented.

**Keywords:** elderly; frailty; immunosenescence; infection; recommended vaccine; comparative effectiveness analysis

## 1. Introduction

The elderly population in the world is increasing rapidly due to an extended life expectancy. Currently, around 8.5% of the world's population (or about 617 million people) are 65 years or older. This number is estimated to continue to increase and reach 17% by 2050 [1]. At that time, one in six people in the world will be elderly. The increased number of elderly in the population provides many opportunities but also poses many health challenges that must be anticipated. In developed countries, the main health problems are chronic non-communicable diseases (cardiovascular disease, diabetes, chronic lung disease, cancer, and Alzheimer's disease), while in low-income countries, the elderly also face the threat of non-communicable and infectious diseases [1].

In the elderly, innate and adaptive immune functions decrease, which is known as immunosenescence [2,3]. Chronic antigen exposure throughout life results in a chronic low-grade inflammation referred to as inflammaging [4,5]. A natural consequence of inflammaging is a decrease in the immune function (immune paralysis) or tolerance of the natural immune system [6]. The relationship between immunosenescence and inflammaging is two-way. Immunosenescence causes inflammaging and vice versa [7]. Both contribute to an increased risk of chronic non-communicable diseases, acute infectious diseases [8], and

frailty. Frailty is a geriatric syndrome that plays a role in increasing a person's vulnerability to infection [9]. Immunosenescence, infection, and frailty are involved in a pathophysiological relationship that influences each other in a vicious cycle of recurring degenerative illness [10].

Infectious diseases are known to cause more than 33% of deaths in the elderly [11]. Immunosenescence and uncontrolled comorbid diseases trigger susceptibility to infection. Infections in the elderly often show atypical clinical symptoms that delay diagnosis. In addition, changes in the pharmacokinetics of antibiotics can result in the ineffectiveness of the antibiotic therapy given. Some infectious diseases such as influenza, pneumococcal pneumonia, and herpes zoster often pose a serious threat to the elderly. Other infections that need to be mentioned are malaria, dengue, and COVID-19, which have been shown to cause higher mortality rates in the elderly population. Those diseases can be prevented with vaccination [11,12]. However, vaccinations in the elderly present their own challenges due to uncontrolled immunosenescence, inflammaging, and comorbid diseases, which will cause a low response to vaccines [13].

Special strategies are needed to improve the vaccine response in the elderly. Several adjustments have been recommended to overcome the immunosenescence process, both in the natural and adaptive immune systems [14,15]. This article aimed to review the published papers which describe several aspects of vaccination in the elderly including frailty, immunosenescence, and inflammaging; decreased immune response and the threat of infections that can be prevented with vaccination; suitable vaccine formulations; and various strategies that can be performed to improve the response to vaccination among the elderly [16–18]. Based on these descriptions and studies, vaccine recommendations for the elderly population will be delivered.

## 2. Frailty, Immunosenescence, and Inflammaging in the Elderly

Aging is characterized by functional decline and deterioration at the cellular, tissue, and organ levels. This wasting causes the loss of homeostasis and a decrease in the ability to adapt to endogenous and exogenous stress, which ultimately leads to increased susceptibility to disease and mortality. Everyone has a different onset and rate of progression in the aging process, which reflects the magnitude of their respective functional capacities. Functional capacity is determined by the ability of cells, tissues, and organ systems to work optimally and is influenced by genetic and environmental factors. The optimal performance of a cell, organ, and individual is maintained through a maintenance mechanism that includes DNA damage repair and protein and lipid synthesis. The process is monitored and maintained for accuracy, with the detection and elimination of defective proteins and lipids, as well as defense against injury and pathogens. These maintenance mechanisms also determine homeostatic conditions through cellular responses (senescence, apoptosis, and repair) and systemic responses (immune system activation or inflammation). DNA damage due to various stressors in the elderly will be responded to by senescence (a halt or a slower level of cell proliferation to allow the repair process) or apoptosis (cell elimination if the damage is too severe to repair). The results are improvement and recovery from the condition (if the repair process goes according to expectations), changes in phenotype (if the repair process is not perfect), or damage and neoplasms (if the repair process goes wrong and is out of control) [9].

In the aging process, the elderly also experience frailty, a geriatric syndrome characterized by weakness and weight loss, which results in low physical activity. Frailty occurs because of the decline in physiological, physical, and mental functions. Frailty manifests as biological susceptibility to stressors and decreased physiological reserves to maintain homeostasis [19]. Both are caused by defects and the failure of repair processes at the cellular, tissue, and organ levels, which are involved in the pathobiology of aging and can lead to chronic disease, multimorbidity, and frailty. Multisystem dysregulation is found in frailty (Figure 1).

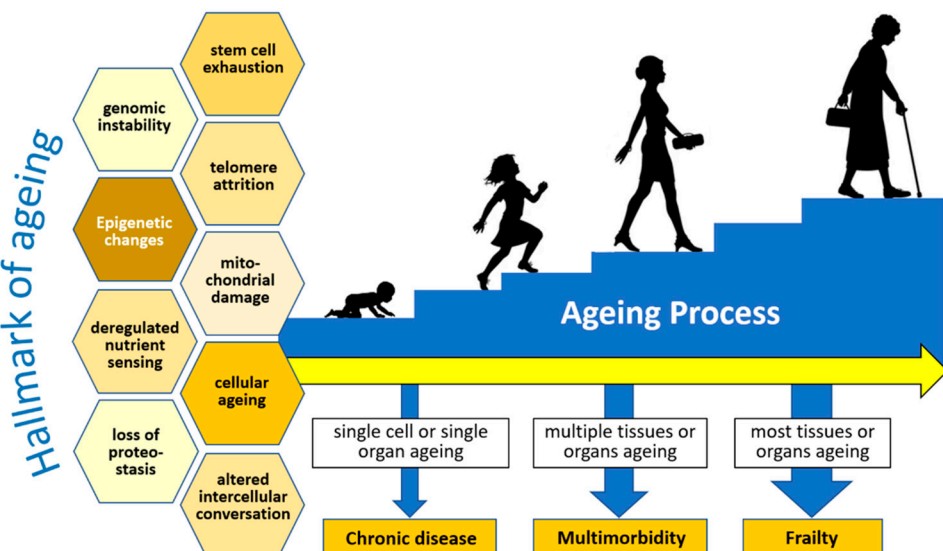

**Figure 1.** The multisystem aging process and frailty. Redrawn and adapted from Thillainadesan, J., et al. [19].

With age, the accumulation of endogenous and exogenous physiological stress will trigger low-level systemic inflammation mediated mainly by immune cells and aged cells. This phenomenon is known as inflammaging. At the same time, the decrease in functional immune cells leads to a diminished ability to fight new bacterial infections, which is called immunosenescence [10]. Declined immune function can be seen in the components of the innate and adaptive immune systems. Phagocytic cells such as neutrophils and macrophages have decreased chemotactic and phagocytic functions in the response to pathogens as well as in the production of reactive oxygen species (ROS) and cytokines. Dendritic cells lose their ability to present antigens and evoke a T lymphocyte response. The response to toll-like receptor (TLR) agonist stimulation is also slowed. The thymus gland undergoes an involution process, and changes in the population and capability of hematopoietic stem cells in the bone marrow result in a decrease in naive lymphocytes, an increase in memory cells, and a decrease in lymphocyte receptors diversity and antibody production. The immune system in the elderly is unable to respond normally to exposure to new antigens and vaccines, making the elderly more susceptible to infections and other health problems, including cancer [2,4].

Decreased immune response and cellular aging lead to low-level systemic inflammation (inflammaging) and immunosenescence. Inflammaging and immunosenescence also trigger a decreased immune response and cell aging [20]. Immune cells in the elderly tend to produce many inflammatory mediators, which weaken their immune function, such as in the case of COVID-19 in the elderly [21]. In immunosenescence, there is an accumulation of senescent cells and a decreased ability to eliminate proteins or defective cells due to the accumulation of cellular debris from cellular damage or death (DAMP). All of these conditions tend to trigger the immune system to produce proinflammatory cytokines, contributing to the inflammatory state. For these reasons, some experts consider inflammaging as the "dark side" of immunosenescence [4].

There is an interconnection between immunosenescence and inflammaging with frailty and infection (Figure 2). Through the formation of some proinflammatory cytokines, such as interleukin-6 (IL-6), C-reactive protein (CRP), and tumor necrosis factor-alpha (TNF-$\alpha$), immunosenescence and inflammaging are important in the pathogenesis of frailty. The elderly who experience frailty with impaired physical and cognitive function are more susceptible to infection with a prolonged clinical course and complications. The infection causes an accumulation of irreparably damaged cells or tissues, which in turn can lead to impaired physical and cognitive functions that lead back to frailty. This interconnection places immunosenescence, inflammaging, infection, and frailty in a pathophysiological pathway that forms a vicious cycle of degenerative disease [10]. It remains a formidable

question whether interventions such as vaccinations for the elderly will be able to break this vicious cycle of aging and illness.

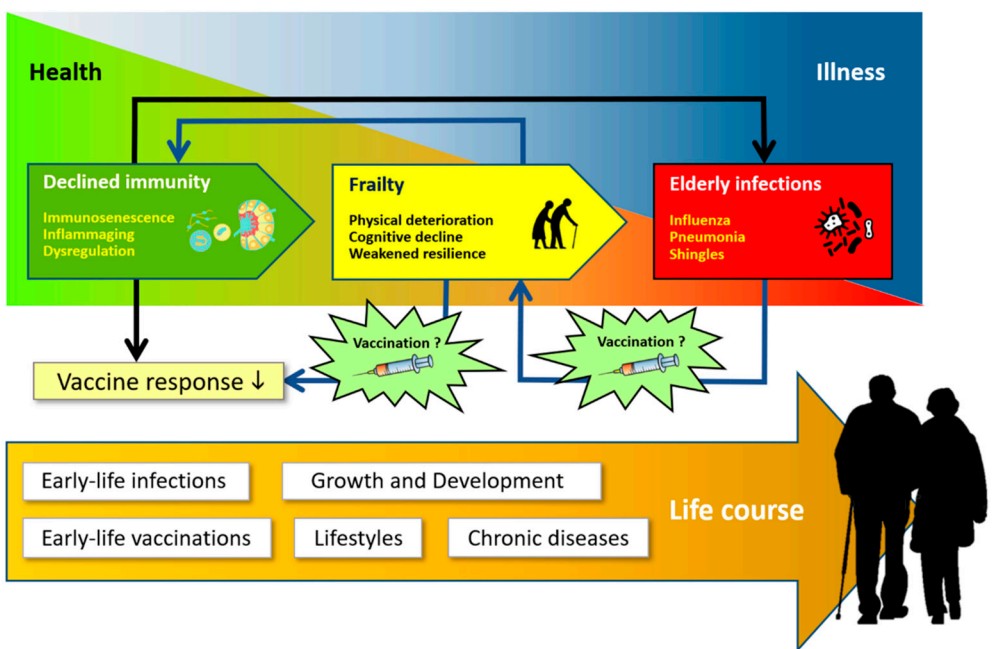

**Figure 2.** Interconnection between immunity, frailty, and infection. Redrawn and adapted from Vetrano, D.L., et al. [10].

### 3. Infection Threat in the Elderly

Immunosenescence, comorbid diseases, and several health-related social conditions (e.g., living in a nursing home, difficult access to health services, etc.) make the elderly more susceptible to infection. Infection is the leading cause of death in 30% of the cases involving elderlies, and it further aggravates underlying disease conditions and reduces functional capacity. Several chronic comorbid diseases such as cardiovascular disease, type II diabetes mellitus, chronic lung disease and other non-communicable diseases increase an elderly person's susceptibility to infection and decrease the response to vaccines. Some infectious diseases that are often found in elderly patients include respiratory tract, urinary tract, skin, and soft tissue infections [11,22]. In terms of the possibility to prevent infection, data from several developed countries such as Australia, Europe, and Canada show that the three most dominant infectious diseases that can be prevented with vaccinations are pneumococcal pneumonia, influenza, and herpes zoster [23–25]. Experience during the pandemic necessitates that COVID-19 must also be added to the list [26].

One method that can be used to display the magnitude of the burden caused by disease is to conduct a disease burden analysis, combining and comparing the impact of several types of disease on a population. This method uses information from various sources to consistently calculate the fatal and non-fatal effects of various diseases, which are combined into a summary of health information referred to as "disability-adjusted life year" (DALY). Disease burden can be estimated using the quantity and quality of life, describing the magnitude and severity of the impact of a disease on the population [24]. According to a survey from Australia in 2015, it was reported that the three diseases with the largest DALYs were pneumococcal pneumonia, influenza, and herpes zoster, especially in the elderly over 65 years [23]. It seems that each country or region has different types of disease burden, which are influenced by its national budget for healthcare financing, different resources, and infrastructures between low-income and middle- or high-income countries, as shown by the results of global studies [27–29] as well as local studies in India [30] and China [31]. As mentioned above, the impact caused by COVID-19 during the pandemic cannot be overlooked [30,32–34].

### 3.1. Pneumococcal Pneumonia

A significant cause of disease morbidity and mortality worldwide is community-acquired pneumonia (CAP). The most common pathogen is *Streptococcus pneumoniae* (pneumococcus), which includes asymptomatic nasopharyngeal carriage, non-invasive pneumococcal pneumonia, and invasive pneumococcal disease (IPD). The Global Burden of Disease Study in 2016 estimated that there were 197 million cases of lower respiratory tract infections caused by pneumococcus with 1–2 million deaths worldwide. The health burden caused by pneumococcal infection is U-shaped, with high mortality found in children under five and the elderly aged 65 years or older. The incidence of pneumonia in the United Kingdom (UK) and the United States of America (USA) has been increasing in those with chronic diseases and in immunocompromised individuals [35]. *Streptococcus pneumoniae* has more than 100 serotypes, which are classified by their capsule polysaccharide composition. The distribution of *Streptococcus pneumoniae* varies depending on the geographical location and the period of the study.

### 3.2. Influenza

Influenza or the 'seasonal flu' is an acute respiratory infection caused by the influenza virus that circulates throughout the world. The resulting health burden can occur throughout the year. In countries with four seasons, influenza epidemics usually occur in the winter season, unlike in tropical countries, where it occurs throughout the year regardless of the season and causes sporadic increases in the number of cases. Influenza can cause illness with a variety of manifestations, from mild colds to severe infections requiring hospitalization and death. The severe cases particularly occur in high-risk populations such as toddlers, the elderly, pregnant women, and individuals with chronic illnesses (such as asthma, diabetes, and cardiovascular disease). Among the four influenza virus serotypes, influenza A and B often infect humans and cause the 'seasonal flu'. Influenza viruses known to be circulating in the world today include two subtypes of Influenza A (A/H1N1pdm09 and A/H3N2) and two subtypes of Influenza B (B/Victoria and B/Yamagata). Data from the Global Burden of Disease Study in 2017 reported that as many as 54.5 million cases of lower respiratory tract infections were caused by influenza. As many as 8.2 million of these infections were serious cases that caused 145,000 deaths [36,37].

### 3.3. Herpes Zoster

Herpes zoster is caused by the reactivation of the Varicella-zoster virus. Primary infection with the Varicella-zoster virus causes varicella. After the disease subsides, the virus remains in the dorsal root ganglia of the spinal nerve. Over time, the Varicella-zoster virus can undergo reactivation and cause a very painful maculopapular rash along one or two adjacent dermatomes called herpes zoster. The most common complication of herpes zoster is postherpetic neuralgia (PHN), which can last for months and sometimes years. The risk of developing PHN increases with age. The elderly tend to experience longer-lasting disease and more intense pain. It is estimated that about 10–13% of individuals over the age of 60 who have herpes zoster will develop PHN. Other complications of herpes zoster include eye complications (herpes zoster ophthalmicus), secondary bacterial infection, cranial and peripheral nerve palsy, and systemic complications such as meningoencephalitis, pneumonitis, hepatitis, and acute retinal necrosis [38].

### 3.4. Coronavirus Disease 2019 (COVID-19)

Since it first broke out in Wuhan, China, in December 2019, COVID-19 caused by SARS-CoV-2 has spread throughout the world. The WHO declared it a pandemic in March 2020 [39]. Globally, as of 29 March 2023, a total number of 761,402,282 confirmed cases of COVID-19, including 6,887,000 deaths, have been reported to the WHO [40]. Since the beginning, collected and reported cases in China have shown that the death rate from COVID-19 increases exponentially with increasing patient age, a tendency that has also been demonstrated in several other regions in the world [41,42]. COVID-19 in the elderly

often causes more severe disease and consequently, higher mortality, with a case fatality rate of around 3% in the 50–59 years age group, around 4–5% in the 60–69 years age group, around 8–11% in the 70–79 years age group, and up to 13–16% in 80 years and older age group [42]. Apart from being caused by the high proportion of elderly subjects who previously had several underlying diseases, such as diabetes, cardiovascular disease, and cancer [43], several observations showed that the immune system of elderly individuals has a higher inflammatory response and a lower ability to overcome infection [44,45]. Weaker and slower response to antigen exposure paradoxically triggers increased levels of interleukin-6 (IL-6), IL-8, and other proinflammatory cytokines, such as IL-1β, IL-1RA, GM-CSF, MCP1, MIP1α, MIP1β, and TNF-α, that can lead to a cytokine storm [45,46]. The consequences can be an ARDS and multi-organ failure, which can end fatally and therefore increase mortality [47].

## 4. Decreased Response to Vaccines among the Elderly

The immune response to vaccines is determined by many factors, including host intrinsic factors (age, sex, genetics, and comorbid diseases), perinatal factors (length of the fetus in the womb, birth weight, breastfeeding method, and maternal factors), extrinsic factors (previous immune status, composition of microbiota in the body, presence of infection, and the use of antibiotics), environmental factors (geographical location, season, the number of family members, and exposure to toxins), and modifiable behavioral factors (smoking, dietary intake, alcohol consumption, exercise, and sleep patterns). In addition, vaccine factors (type of vaccine, vaccine product, adjuvant, and vaccine dose) and the method of administration (vaccine schedule, injection site, route of administration, and concomitant drugs) also determine the immune response to vaccines [48].

As previously described, the immune system in the elderly undergoes a change called immunosenescence [13]. In addition to developing an increased risk of infection, immunosenescence also lowers the protection after vaccination. Immunosenescence affects the function of innate and adaptive immune cells which include preventing the effective induction of memory lymphocytes and reducing the effect of booster vaccinations. This lowers the antibody response in the elderly and causes the antibody titers to wane more rapidly. Therefore, the protective effect of vaccination cannot be fully guaranteed in the elderly [13,16,49]. Changes in the response to vaccines in the elderly are summarized in Figure 3 [49].

The most influential component in decreasing the immune response to vaccines is thymic involution. Additionally, this is accompanied by shifts in the proportion of lymphoid progenitors in the bone marrow. The aging process causes a relative increase in the proportion of myeloid cell progenitors, while the proportion of lymphoid cell progenitors decreases. Meanwhile, thymic involution and atrophy cause a decrease in naive T cell output by 3% per year. Homeostatic balance in peripheral lymphocyte cell groups is maintained by increasing the number of mature T cells in the periphery, leading to an increase in the ratio of memory T cells to naive T cells. However, lymphocytes cannot proliferate indefinitely. In the elderly, the proliferative capacity of lymphocytes decreases due to telomere shortening. Telomeres shorten during each cell division cycle (about 50–100 bp per division cycle). When they reach a critical limit, the short telomeres will stop the ability of lymphocytes to divide. What remains in the lymphocyte are "old" T cells that are no longer able to proliferate and have limited receptor diversity. In addition, they can no longer express the co-stimulating molecule CD28 but instead express killer immunoglobulin-like receptors (KIR), the NKG2D receptors, and lymphocyte function-associated antigen 1 (LFA-1). These changes in T cells will in turn lead to decreased stimulation of B cells to differentiate into plasma cells that produce adequate antibodies [49].

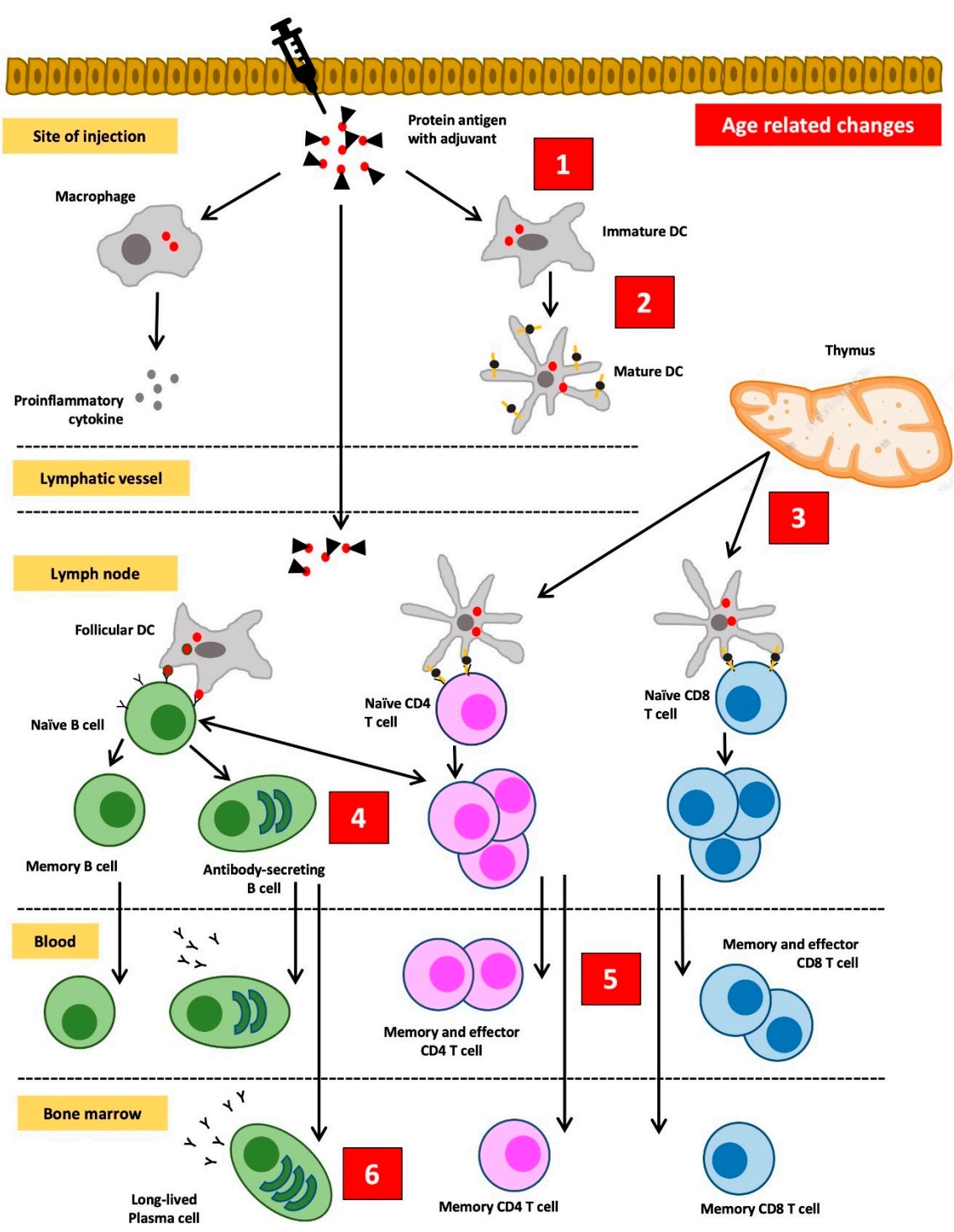

**Figure 3.** Changes in the response to vaccines in the elderly. (**1**) Subclinical inflammation increases the threshold of „danger" signals to be induced by vaccines augmented with adjuvant. (**2**) Functionally defective antigen-presenting cells will interfere with the uptake and the presentation of antigens. (**3**) Involution of the thymus gland leads to the advanced loss of naïve T cells, thus compromising the success of primary vaccination. (**4**) Reduced B cell production and faults in isotype switching and somatic hypermutation result in weak and low-affinity antibody responses. (**5**) An increased number of effector T cells limits immune cell diversity and threatens the desired effects of vaccination. (**6**) The aging process in the stroma of the bone marrow can compromise the viability of plasma cells and shorten the duration of immunological protection. Redrawn and adapted from Weinberger, B., et al. [49].

Evidence of a decreased response to vaccination can be seen in various studies that have been conducted on both animals and humans. Studies in rhesus macaques show a lower antibody response in older macaques [50]. Following influenza vaccination, antibody titers in young adults were consistently higher than in the elderly. The risk of being infected with influenza after vaccination occurs in the elderly who do not show humoral or cellular immune response [51]. In a study evaluating changes in the repertoire of B cells and IgM and IgA antibodies after pneumonia vaccine administration, IgA and IgM responses were found to be significantly lower in the elderly [52]. The quality of the antibodies produced was also inferior to the group of young subjects [53]. The efficacy of the herpes zoster vaccine also significantly decreased in the elderly group, reaching 41% in the >70-year age group and only 18% in the >80-year age group [54]. Thus, appropriate strategies are needed so that the elderly can obtain higher and more effective vaccine protection.

## 5. Strategies to Increase the Vaccine Response in the Elderly

There are several strategies that can be pursued to improve the response to vaccination in the elderly. These strategies include: improving vaccine formulations, using higher antigen concentrations, using more immunogenic adjuvants, giving booster injections, choosing the proper method and timing of vaccine administration, adjusting immunization schedules, alternative routes of vaccine administration and injection sites, and encouraging the use of senolytic drugs or immunomodulators [55].

### 5.1. The Use of Higher Antigen Doses

It is necessary to formulate a new vaccine that is adapted for the elderly population. One of the strategies that can be used is to increase the dose of the antigen. It has been shown in several studies using multiple vaccines that higher doses of antigen will result in higher and more protective antibody response, as seen in the Fluzone® trivalent [56] and quadrivalent [57] influenza vaccines. However, the same results cannot be expected for all types of vaccines. The number of antigens per virion contained in the vaccine is more decisive than the total number of virions.

### 5.2. The Use of More Immunogenic Adjuvants

Adjuvants are substances added to enhance the immune response to a vaccine. The magnitude and quality of the response to the vaccine are highly dependent on the type of adjuvant added to the vaccine. The first adjuvant used in vaccines for humans was aluminum salts, which have been used since the 1920s. However, the use of aluminum is often not sufficient to increase the immunogenicity of vaccines in the elderly.

A stronger adjuvant is needed to overcome this limitation. Most research on adjuvants has focused on influenza vaccines to achieve one of the following goals: the induction of higher protective antibody titer, the introduction of a wider range of antibodies against a variety of epitopes that allow cross-protection against viral strains, the use of lower doses of antigen, as well as the induction of protective mucosal immunity [58]. Some promising candidates include oil-in-water emulsion adjuvant MF59 [59] and AS03 for influenza vaccine, AS04 for hepatitis B, and AS01 for Varicella-Zoster vaccine [60]. Several other types of adjuvants that are being developed and investigated include QS21, CpG oligodeoxynucleotide, and flagellin [58,60,61]. The second-generation Herpes-Zoster vaccine adjuvant, Shingrix®, has been shown to exhibit high immunogenicity and efficacy in the elderly and provides protection for up to 9 years [62].

### 5.3. Delivering Booster Injections

Antigen exposure in primary vaccinations will activate naive T cells to differentiate into effector T cells and memory T cells. Similarly, B cells will be activated by T cells to differentiate into plasma cells that produce antibodies and memory B cells. Booster vaccinations will trigger memory T cells and B cells to respond rapidly and stronger. The facts show that the elderly are not able to maintain long-term memory responses. Antibody

titers decline more rapidly in the elderly and fail to reach protective levels [50]. Animal studies also support these findings. The antibody response to the primary vaccine was significantly decreased in old rhesus macaques; however, giving booster injections at an interval of one month after the primary vaccine achieved a response equivalent to that in young adult macaques [63]. This kind of booster strategy has never before been applied to other types of vaccines that exist. The annual influenza vaccine is given only once a year. The booster for the *Streptococcus pneumoniae* vaccine is also only given once after 1 year or 5 years. It is necessary to conduct research on whether giving a second booster (especially for inactivated vaccines) will be able to increase the antibody response in the elderly.

The experience from vaccination during the COVID-19 pandemic [64] has led experts who are members of The European Technical Advisory Group of Experts on Immunization (ETAGE) to recommend additional booster injections for susceptible immunocompromised groups, including the elderly [65]. The same step was taken by the UK government [66] at an interval of 6 months after the previous injection of the COVID-19 vaccine. Giving booster vaccine injections appears to be necessary for long-term protection in the elderly, and its efficacy is also determined by the presence of plasma cells and memory B cells produced by the primary vaccination [67]. Primary vaccinations should be scheduled and completed at a young age to obtain a better response to booster injections.

### 5.4. Exploration of Alternative Routes of Vaccine Administration

The route of antigen administration has an important role in triggering protective immunity. Adjustment in the vaccine route administration is one of the strategies to improve the response of the elderly to vaccines. In the past, the preferred route of vaccine administration was a percutaneous injection, including both subcutaneous and intramuscular injections. Recently, microinjection techniques have been developed that allow the administration of vaccines with the intradermal route [68]. In the skin, there are many immune cells (dendritic cells and macrophages) that can act as antigen-presenting cells. These cells will migrate to nearby lymphoid glands, activate antigen-specific T cells, and stimulate the differentiation of follicular helper T cells in the lymphoid glands. Follicular helper T cells will in turn induce antigen-specific B cells and their differentiation into antibody-producing cells. Intradermal administration of vaccines theoretically would facilitate antigen uptake by Langerhans cells, thus allowing the use of lower doses of antigen to induce a protective immune response in the elderly [69,70]. Vaccines given with the intradermal route give rise to higher immunogenicity and seroconversion rates compared to the intramuscular route, although they also cause more local reactogenicity at the injection site [71,72]. Problems that must be considered when using intradermal vaccinations in the elderly are changes in physiology, vascularity, and skin integrity, including a decreased number of antigen-presenting cells in the elderly skin.

Another route under investigation is the mucosal administration of vaccines, particularly for respiratory infections such as influenza and pneumonia [73]. The number of antigen-presenting cells in the mucosal tissue will increase the response to the antigen. Antigen exposure through the mucosa will trigger antigen-specific T cells and IgA+ B cells, which then will migrate to lymphoid glands, enter the blood circulation, and finally fill up the mucosal tissue. Entering the mucosal tissue, lymphocytes will differentiate into effector cells and memory cells. Administration of the vaccine through the nasal mucosa will produce an IgA response in the mucosa and an IgG response systemically [74]. Research on mucosal vaccines, specifically for the elderly, is still very limited.

### 5.5. Vaccine Injection Time

The immune system is also affected by circadian rhythms, and the immune response varies at different times of the day. Cytokine and hormonal diurnal rhythm patterns also determine a person's response to vaccines [75]. Several studies using influenza vaccines and COVID-19 vaccines have shown that the administration of a vaccine in the morning triggers better antibody titer than the administration in the afternoon [76,77].

*5.6. The Use of Senolytic and Immunomodulatory Drugs*

It is known that in the elderly, there is an accumulation of senescent cells in various tissues. T cells in the elderly show decreased T cell receptor diversity and impaired signaling. Memory T cells have defects in double-stranded DNA and telomere shortening. B cells in the elderly have several defects, including decreased ability of class switch recombination, decreased somatic hypermutation, decreased capacity of antibodies to bind and neutralize antigens, decreased specificity of antibody binding, epigenetic changes that correlate with a decreased antibody response, and an increased number of inflammatory B cell subsets. All of these defects contribute to a decreased response to vaccinations. Recently, several pharmacological strategies are being developed to modulate the immune system or eliminate the population of senescent cells in the elderly to enhance the immune response. Senolytic drugs selectively induce the apoptosis of senescent cells without disturbing the population of healthy cells. Its potential utilization has been tested in preclinical studies and is promising, including the use of the mTOR inhibitor everolimus [78,79] or the autophagy activator spermidine [80]. The first senolytic drugs developed include Dasatinib, Quercetin, Fisetin, and Navitoclax [81].

## 6. Recommended Vaccines for the Elderly

Current vaccine recommendations for the elderly include vaccines for influenza, pneumococcal pneumonia, COVID-19, Herpes-Zoster, and booster vaccines for tetanus and diphtheria. Although vaccines can prevent severe infections in the elderly, in general, the protection provided is lower than in the young adult population. To increase the effectiveness of vaccines in the elderly, several efforts have been made as previously discussed. In recent years, several new types of vaccines have been successfully developed, i.e., vaccines for dengue and malaria. As is known, in many tropical countries, such as those in the Caribbean (including Puerto Rico), Central and South America, Southeast Asia, and the Pacific Islands, dengue is still a dangerous threat, especially for the elderly. In addition, malaria remains a major health problem in tropical and subtropical areas, especially in African countries and some parts of Oceania such as Papua and Papua New Guinea.

*6.1. Influenza Vaccine*

The vaccine for seasonal influenza currently licensed for distribution is a trivalent or quadrivalent inactivated split virion vaccine. The vaccine contains antigens from two subtypes of the Influenza A virus and one or two circulating Influenza B virus strains. The composition of seasonal influenza vaccines is updated annually based on data from epidemiological and virological surveillance. The surveillance conducted by the Global Influenza Surveillance and Response System (GISRS) was established by the World Health Organization (WHO) to monitor influenza virus activity worldwide.

Older adults ($\geq$65 years) are considered a priority group for annual influenza vaccination due to high influenza-related morbidity and mortality each year. To overcome the low effectiveness of the vaccine, the recommended vaccines are enhanced inactivated influenza vaccines (eIIV) which include Fluad® (MF59-adjuvanted, A-eIIV), Fluzone® (high dose quadrivalent vaccine, H-eIIV), and Flublok® (recombinant HA, R-eIIV). Each of these eIIV had been shown to induce higher hemagglutinin inhibition (HI) titers in the elderly and give out superior immunogenicity and/or efficacy in preventing medical complications related to influenza against its comparator. A systematic review and meta-analysis from 1993 papers reported that the Fluad® vaccine showed a better protective effect against hospitalization related to pneumonia (adjusted risk ratio 0.75 (95% CI: 0.57–0.98)) and cases of influenza (confirmed by a laboratory) (adjusted odds ratio 0.37 (0.14–0.96)) compared to vaccines without adjuvants [82]. The addition of MF59® adjuvants has been shown to increase antibody production (13–14% relative antibody increase, with Geometric Mean Titre (GMT) ratio of 1.43 ($p < 0.001$)), increase seroconversion and seroprotection (8–14% relative increase in the percentage of subjects with HI titer $\geq$ 160), and increase antibody binding affinity, causing wider diversity of antibody epitopes and wider serological protection

against mutated strains. The high-dose Fluzone® quadrivalent vaccine contains four times the antigen as a standard-dose vaccine and has been shown to have better immunogenicity in the elderly. In a large, randomized, phase IIIb-IV, multicenter study involving around 32,000 subjects aged 65 years or older in the USA and Canada, the high-dose vaccine showed an advantage in preventing laboratory-confirmed influenza (relative efficacy of 24.2% (95% CI: 9.7–36.5)) versus the standard-dose comparator [83].

Flublok® is a recombinant hemagglutinin (rHA) vaccine. This vaccine is a purified product, containing three times the amount of hemagglutinin in standard vaccines, which is typically around 15 g of recombinant hemagglutinin per strain. The rHA vaccine does not contain neuraminidase and is not contaminated with chicken egg protein. In a randomized clinical trial with a control group, the rHA vaccine triggered HI antibodies with a geometric mean titer that was significantly higher than the standard vaccine (338.5 (299.7–382.5) vs. 199.2 (176.8–224.4)). Seroconversion to the influenza A antigen was also higher in the group receiving rHA (43% vs. 33%, $p = 0.001$). More convincing results were obtained in the age group ≥75 years [84]. High-dose influenza vaccines, adjuvant vaccines, and rHA vaccines generally have a good safety profile and are well-tolerated. In a randomized controlled trial study comparing head-to-head the four types of influenza vaccine available to the elderly: the standard-dose quadrivalent vaccine, trivalent vaccine with adjuvant MF59, high-dose trivalent vaccine, and recombinant hemagglutinin quadrivalent vaccine, it was found that elderly people receiving the "enhanced vaccines" had better humoral and cellular immune responses compared to the elderly who received standard-dose vaccines [85].

Vaccine formulation also determines the magnitude of the immune response, particularly the cellular immune response. Currently, licensed influenza vaccines are generally produced using either split-virus or subunit formulations. Subunit vaccines undergo multiple purification steps, unlike split-virus vaccines. Consequently, split-virus vaccines contain higher amounts of viral internal proteins (e.g., matrix M1 protein and nucleoprotein). Internal proteins of the influenza virus are important for stimulating cytotoxic T cells (CD8+). Studies comparing split-virion influenza vaccine versus subunit vaccine in the elderly found 77.8% (95% CI: 58.5–90.3%) efficacy for the split-virion vaccine compared to 44.2% (95% CI: 11.8–70.9%) for subunit vaccine [86].

### 6.2. Pneumococcal Vaccine

Two types of pneumonia vaccines currently available are the 23-valent polysaccharide vaccine (PPSV-23) and the 13-valent conjugate vaccine (PCV-13). The PPSV-23 vaccine provides protection against serotypes 1, 2, 3, 4, 5, 6B, 7F, 8, 9N, 9V, 10A, 11A, 12F, 14, 15B, 17F, 18C, 19F, 19A, 20, 22F, 23F, and 33F. The PCV-13 vaccine provides protection against serotypes 1, 3, 4, 5, 6A, 6B, 7F, 9V, 14, 18C, 19A, 19F, and 23F. The PPSV vaccine contains only polysaccharide antigens that do not trigger a T cell immune response and thus do not produce adequate antigen-specific memory B cells. In the elderly, the response to PPSV-23 is lower than in young adults and decreases over time. For example, vaccine effectiveness (VE) in preventing pneumococcal pneumonia was only 24% (95% CI: −6 to 46) in older adults compared to 32% (95% CI: −21 to 62) in younger subjects. For preventing vaccine-type pneumonia, VE in older adults was only 28% (95% CI: −10 to 53) effective compared to 40% (95% CI: −6 to 69) in the younger counterpart [87].

In a population study conducted by British Public Health in England and Wales on the elderly aged 65 years who received PPSV-23 vaccination during 2012–2016, vaccine effectiveness against Invasive Pneumococcal Disease (IPD) was calculated as 100% × (1 − the odds of vaccination in vaccine-type IPD cases/odds of vaccination in non-vaccine-type IPD controls) using logistic regression to adjust for potential confounding factors. It was found that the effectiveness of the vaccine decreased with time and with increasing recipient age (31% (95% CI, 16–44) in the 65–74-year age group and to 17% (95% CI: −3 to 32) in the 75–84-year age group). Comparing VE by risk group showed that the lowest VE was in the immunocompromised group [88]. The PCV-13 vaccine was later introduced and is currently the first choice for the elderly in many countries. This

vaccine has better immunogenicity than the polysaccharide vaccine. Conjugation with protein makes it able to stimulate CD4+ helper T cells, which trigger the emergence of polysaccharide-specific memory B cells [89].

The PCV-13 vaccine, which is used as a primary vaccine (priming dose) in the elderly, will trigger higher titer antibodies and is more effective than the PPSV vaccine as a primary vaccine injection [90]. In the CAPITA study involving 84,496 elderly people aged 65 years, PCV-13 was shown to reduce the need for hospital care due to vaccine-type strains of community-acquired pneumonia by 45.6% and the incidence of invasive pneumococcal disease by 75% [91]. PCV-13 activates a stronger and longer-lasting immune response than PPV-23, but PPV-23 contains more viral serotypes. Each country publishes different vaccination guidelines for the elderly. The United States Centers for Disease Control and Prevention (US-CDC) provides the following recommendations [92]:

For elderly ≥65 years who have never received a pneumococcal vaccine before, it is recommended to get the PCV-15 or PCV-20 vaccine (note: both are not yet available in Indonesia). If PCV-15 is selected, then at least the following year should be followed by the PPSV-23 vaccine at a minimum interval of 8 weeks. The same recommendation may also be considered for adults with immunocompromised conditions, cochlear implants, or cerebrospinal fluid leaks. If PCV-20 is selected, a booster injection with PPSV-23 is no longer required.

- For elderly ≥65 years who have previously received the PPSV-23 vaccine, it is recommended to receive one dose of the PCV-15 or PCV-20 vaccine at least a year after the previous PPSV-23 injection. There is no longer a need for a booster shot with PPSV-23.
- For elderly ≥65 years who have previously received the PCV-13 vaccine, it is recommended to get the PPSV-23 or PCV-20 vaccine (if PPSV-23 is not available)
- For immunocompromised adults that reach 65 years and have previously received the PCV-13 vaccine followed by the PPSV-23 vaccine before 65 years, it is recommended to get a PPSV-23 booster vaccine with an interval of at least 5 years from the previous PPSV-23 injection.

*6.3. COVID-19 Vaccine*

This is the first time in history that multiple vaccine platforms are available for the same target virus, SARS-CoV-2, particularly targeting the spike protein that is used by the virus to gain access to our cells by binding to ACE2 receptors. These include whole-inactivated virus vaccines, viral (adenovirus)-vectored vaccines, protein subunit vaccines, virus-like particle vaccines, and nucleic acid vaccines (DNA and mRNA vaccines) [93]. An unprecedented convergence of effort from scientists, governments, financing resources, and pharmaceutical companies on one viral infection, and the use of technology previously aimed at developing vaccines for HIV, MERS, or Ebola, such as reverse genetic technology, has made it possible to construct a synthetic SARS-CoV-2 mRNA sequence that encodes the spike protein [94]. This technique greatly accelerates the manufacture of vaccines and induces a more robust immune response.

Based on experience and studies on the ongoing global COVID-19 vaccination, there are several facts that can be noted: (i) antibody responses to vaccines are lower in the elderly and in those who have comorbidities such as hypertension, uncontrolled diabetes, obesity, or smoking [95]; (ii) antibodies from vaccination (regardless of the vaccine platform) only last for a short time period and on average decrease after 5–6 months [96,97], so giving booster injections is considered necessary; (iii) the constantly changing circulating SARS-CoV-2 variants, due to its tendency of mutations with the ability to escape the antibody response, necessitates revaccination with a suitable vaccine [98,99].

The elderly are a top priority for getting the COVID-19 vaccine because data show that their morbidity and mortality are significantly higher than in the younger age group [42,43]. The preferred platform is one that is proven to have a high vaccine efficacy such as the mRNA vaccine (mRNA-1273 or BNT162b2 vaccine). Most of the primary doses of available vaccines were given in two shots with variable intervals [100]. With regards to the

booster vaccine, some studies indicated that heterologous regimens significantly induced stronger immunogenicity and reactogenicity compared to homologous regimens with tolerable adverse effects [101–103]. Levels of neutralizing antibodies were also higher in the heterologous booster participants who received the m-RNA vaccine as booster [103]. Adenovirus-vectored vaccines are known to stimulate humoral and cellular immune responses, while mRNA-based vaccines tend to induce higher levels of antibody to spike protein [100]. This tendency might be related to the difference in antigen presentation and modes of action of both vaccine types [104]. More importantly, according to a recent systematic review and meta-analysis, giving multiple COVID-19 vaccinations to the elderly is considered effective and safe [105].

Even though there are currently no age-based recommendations for the COVID-19 vaccine, given the above-mentioned facts and considerations, it can be suggested that the elderly should have vaccine priority. Vaccination hesitancy among older adults should be handled. The preferred vaccine platform is the mRNA type. Controlling the existing comorbid diseases is very important to get the maximum benefit from vaccination. A booster injection using homologous or preferably heterologous vaccines should be considered after 6 months from the primary vaccine injection. A newly developed bivalent COVID-19 vaccine can also be considered. To anticipate the emergence and the threat of new SARS-CoV-2 variants in the future, we need to develop a pan-coronavirus vaccine or use the influenza vaccination strategy, in which people receive boosters on a periodic basis against the dominant circulating SARS-CoV-2 variant [106].

### 6.4. Herpes Zoster Vaccine

Activation of latent Varicella-Zoster virus (VZV) infection can occur at any time during life. However, factors such as age and conditions associated with decreased T cell immunity increase the risk of activation. The live attenuated Varicella-Zoster virus (Zostavax®) vaccine has received approval to be used to enhance specific cellular immunity against VZV. The Shingles Prevention Study showed that the vaccine could reduce the incidence of herpes zoster by 51.3% and the incidence of PHN by 66.5%, but its efficacy decreased from 63.9% in the 60–69-year age group to 41% in the 70–79-year age group and was even lower in the older age group. In addition, there was also a significant decrease in antibody response over time, specifically 4–8 years after vaccination [54]. This vaccine should not be given to patients with HIV/AIDS or other immunodeficient conditions.

A new recombinant vaccine with adjuvants (Shingrix®) has recently been approved to be used for the prevention of herpes zoster in the elderly. This vaccine uses a liposome-based AS01B adjuvant system added to VZV recombinant glycoprotein E [107]. With this formulation, the vaccine enhances vaccine immunogenicity by stimulating the innate immune system at the injection site and enhancing antigen presentation. This stimulation induces strong Th2 humoral and Th1 cellular immune responses and cross-presentation to CD8+ T cells triggering an early IFN-γ response [108]. Evidence for the efficacy of recombinant vaccines with adjuvants was obtained from two phase III clinical trials, in which two doses of vaccine resulted in protection against herpes zoster by 97% in adults 50 years of age [109] and by 89.8% in the elderly ≥70 years [110]. A strong antibody and CD4+ T cell response were maintained up to 3 years after vaccination [111]. The recombinant subunit of the vaccine with adjuvants has been shown to be superior to the live attenuated VZV vaccine despite the greater risk of an adverse reaction at the injection site [112]. Another advantage of this vaccine is that it can also be used in immunocompromised or immunodeficient patients, such as patients with HIV [113] and transplant patients [114].

### 6.5. Diphtheria and Tetanus Vaccine

Antibody responses to diphtheria and tetanus vaccines are lower in the elderly. The duration of protection is shorter, and a second booster injection after 5 years from the first injection does not cause long-lasting enhancement of immunity [115]. A cross-sectional study in the USA showed that only 47% of adults aged ≥20 years had resistance to both

diseases, and only 63% of adults with tetanus immunity also had protective antibodies against diphtheria [116].

In the USA, the CDC recommends that all adults who never received the tetanus or diphtheria vaccine should get one shot of the diphtheria–tetanus toxoid (Td) vaccine or the diphtheria-tetanus toxoid–acellular pertussis (Tdap; Adacel® or Boostrix®) vaccine. The vaccine injects intramuscularly and should be repeated with a booster injection every 10 years. If possible, for the elderly ≥65 years, it is recommended to give the Boostrix® vaccine [117].

*6.6. Other Vaccines*

Apart from the vaccines discussed above, several new vaccines aimed at preventing dengue virus infection and malaria have been successfully developed. Three types of dengue vaccines that have been licensed or have undergone phase III clinical trials are Dengvaxia® (Sanofi Pasteur), Qdenga® (Takeda), and TV003/TV005 (NIAID/Butantan/Merck), but these three vaccines are primarily intended for pediatric or adolescent patients, and currently, no research data nor any recommendations exist for the elderly group [118]. The same is true for the malaria vaccine. The first malaria vaccine recommended by the WHO is RTS,S/AS01, which is marketed as Mosquirix®, and for now, it is only intended for children in sub-Saharan Africa and in other regions with moderate to high *P. falciparum* malaria transmission; it is not intended for the elderly [119]. It will take some time before we can develop a similar vaccine that is safe and effective for the elderly in the future.

## 7. Concluding Remarks

Currently, the available vaccines have provided some protective immunity in the elderly. However, despite significant successes, the vaccines are still unable to provide long-term protection. Current vaccination strategies should be able to induce long-term antibody responses by enhancing cellular and mucosal immunity. Among the elderly, responses to vaccination vary widely depending on several factors including their baseline immunological profile. Excessive inflammation is detrimental by potentially inhibiting the immune response to vaccines. A new approach is necessary to suppress inflammation so that the response to vaccines among the elderly can be improved. Such an approach includes the development of new vaccine adjuvants specifically stimulating the desired immune response, a higher dose of vaccine, repeated heterologous booster injections, and exploring alternative routes, specific formulations, or the platform of vaccine that more robustly induce the immune response and provide immunological memory that leads to long-term immunological protection or by alleviating the senescence cells that hamper the immune response. Current vaccine recommendations for each vaccine-preventable disease in the elderly are made to address the specific problems of each type of vaccine available, the host's condition, and the characteristics of the disease to be prevented. Further studies are warranted to understand the role of immunosenescence and inflammaging to develop more effective vaccines for the elderly.

**Author Contributions:** Conceptualization, G.S. and D.P.; methodology, G.S.; validation, D.P.; writing—original draft preparation, G.S. and D.P.; writing—review and editing, G.S. and D.P.; visualization and drawing, G.S. and D.P.; supervision, D.P.; project administration, G.S. All authors have read and agreed to the published version of the manuscript.

**Funding:** This research received no external funding.

**Institutional Review Board Statement:** Not applicable.

**Informed Consent Statement:** Not applicable.

**Data Availability Statement:** Not applicable.

**Conflicts of Interest:** The authors declare no conflict of interest.

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
