# Peer review of "Challenges in the Vaccination of the Elderly and Strategies for Improvement"

_pathophysiology, doi:10.3390/pathophysiology30020014_

Round 1

Reviewer 1 Report

This work deals with the challenges of vaccination for the elderly and strategies for Improvement.

Although the manuscript is well written and presented, there are two  points which must be improved before its acceptance for publication.
1- First of all, the authors must indicates if it is a survey paper. Also they must add in the abstract the principal goal of this work as well as the main results and conclusions.
2- The authors must improve the introduction section by adding a paragraph which describes the organization of the manuscript.
3-  Other diseases as malaria and dengue fever have a induced mortality which increase with the age of individuals. What about these two diseases and the development of vaccines?

Reviewer 2 Report

The manuscript revisits immunity changes in the elderly, mainly related to immunosenescence, inflammaging, and frailty, and, consequently, ways it is (negatively) affecting an immune response to vaccination to selected infectious diseases. Applied to influenza, herpes zoster, and pneumococcal pneumonia, authors describe possible strategies to improve the vaccination effect on elderly immunity reaction, such as more immunogenic vaccine formulations, higher antigen dosing, and more potent vaccine adjuvants, recombinant vaccines, booster shooting, immunomodulation, or various time administration of vaccination.

In general, the manuscript form and content are more or less acceptable; however, there are several issues that authors must address to improve the manuscript quality before one can start considering the manuscript's acceptance for publication.

(i) Firstly, one of the most significant of the current version of the manuscript is that the key points the review rely on are a bit outdated. That being said, more specifically, the authors claim that diseases that elderlies could suffer from but are vaccine-preventable are pneumococcal pneumonia, herpes zoster, and influenza. To support this, they use the following reference,

Australian Institute of H, Welfare. The burden of vaccine preventable diseases in Australia: Canberra, 2019, p. 152, which comes from the year 2019 but analyzed data from 2016 (!).

Thus, the authors' endpoints come from the pre-covid era! I do not want to claim that covid-19 pandemia had to change everything regarding vaccination policies necessarily, but at least, the covid-19 belongs to diseases that are vaccination-preventable in the elderly, see

https://doi.org/10.1038/s41564-022-01166-0

and many others. Since the vast majority of references cited in the manuscript come from the year 2020 or earlier, the manuscript seems to me as if it has been "hidden in a drawer" for a long time after the first rejection somewhere and is now just resubmitted with no significant (but necessary!) updates. The covid-19 is mentioned shortly in the manuscript (in sections 5.3. and 5.5.); however, since it has become one of the model infectious diseases, its factual omitting in the manuscript's storytelling seems infeasible. So, the authors have several options for how to address the issue,

(i-a) If they want to avoid covid-19 at all, the manuscript's title, abstract, and introduction have to be rewritten to clearly state that the review is -- for some (obscure) reason or another -- not covering covid-19 and other vaccination-preventable infectious diseases in elderly. This might be really tricky, though, since the covid-19 is likely a game-changer in vaccination policies, not only for the elderly.

(i-b) If authors would decide to consider covid-19 as a vaccination-preventable disease in the elderly, they should update the manuscript so that covid-19 is listed as another disease together with pneumococcal pneumonia, herpes zoster, and influenza.

Even more, other diseases could be preventable by vaccination in the elderly, such as pertussis; the authors should consider that, see https://doi.org/10.3390/vaccines10050641 and others.

(ii) As already mentioned, most references cited by the manuscript come from years before 2020. Considering the recency of the manuscript's topic and its genre as a review, such selective ignoring of whatever news the last three years brought, particularly covid-19, makes the manuscript outdated. Authors should revisit the references and list a significant proportion of current publications on the topic. Vaccination strategies and policies have changed since covid-19; consider non-vector mRNA vaccination, vaccination of elderlies based on fast preclinical studies only, fast designing of vaccination molecules based on chemoinformatics, etc., based on Physiology journal aims and scopes' barriers, of course. This is related to my point (i), too.

(iii) I miss a more quantitative kind of results in many parts of the manuscripts. The manuscript's genre, review, implies there should be hard-fact-based information supported by sufficient scientific evidence and reported by common numerical epidemiological metrics such as odds ratios, relative risks, etc., that the cited original articles have published initially. For example, In 6.1. section, Influenza vaccine, more numerical results comparing various approaches should be listed, e. g. on lines 381--383, how much has antibody production, seroconversion, seroprotection, etc., increased by adjuvant adding, based on the cited studies [62] and [63]? In 6.2. section, Pneumonia vaccine, on lines 416—417, how much lower is a response to PPSV-23 in the elderly than in young adults? Use reference [66] to quantify the effect size and confidence level. The same for lines 418—421, "the lowest effectiveness of the vaccine in aged > 85 years"; how is the effectiveness measured? Please find it in [67] and report it. Same for line 423 and many other many examples.

(iv) Figure 1 seems to be copied as-is from https://doi.org/10.1093/ageing/afaa112. Figure 2 is copied and modified a bit from https://doi.org/10.1016/j.arr.2021.101351. Figure 3 is reproduced as-is from https://doi.org/10.25816/5ebcb64ffa7e3. Figure 4 is copied as-is from https://doi.org/10.1086/529197. The sources are mentioned in figure captions, but there has yet to be claimed that figures are taken as-is from original sources. Still, if so, there are better approaches to research we could emphasize. This is, instead, a potential plagiarism issue (!). Authors might feel free to get inspired by the original figures but should add something new to them and modify them substantially. Where could the manuscript contribute if all figures explaining the paper's hardstone ideas were just copied and pasted? Please ensure the figures are different enough.

(v) Although a missing discussion section could not be a problem in the given manuscript genre, the conclusion seems to be only half-baked. It should adequately highlight and conclude what diseases, why, and how exactly are prevented by vaccination in the elderly. Key findings should be reported using numerical results, if at least a bit possible, following e. g. a form "vaccination boosted by adjuvants increases antibody titers 3.7-times (95% CI 2.5, 4.2) compared to adjuvants non-usage, as proven by all mentioned studies" and so on. The language of the current conclusion is simple enough for an academic text. Please improve it.

(vi) Throughout the manuscript text, there are many typos, hard-to-follow and weird sentence formulations, missing commas and full stops, incomplete sentences, sentence fragments, and many other language issues, both content and formal. For instance, in lines 15--17, the sentence's wording

"Vaccine preventable diseases that threaten the elderly are common cold, 15 herpes zoster and pneumococcal infection, which because life lost in years due to major disability 16 in aging people"

is hardly understandable. On line 23, the last sentence of the abstract is missing a full stop and needs to be completed. Article "a" should not be at the end of the previous lines but at the beginning of the following lines (line 242). And many others. The authors should at least double-check the manuscript's language.

Round 2

Reviewer 1 Report

This version of the manuscript is well improved.

Reviewer 2 Report

Dear authors,

thank you for addressing all my comments and suggestions. I understand some of the suggestions required a lot of invested time and, moreover, a switch of way how to think about the information you worked with. You have improved your manuscript significantly. I believe that at the moment, a broad audience of professionals in clinical or theoretical medicine and pathophysiology and other audience of Pathophysiology journal would find it interesting since it brings valuable contributions and comparisons for practicing and other ongoing research in the field. I appreciate you have updated your manuscript, added a section dedicated to COVID-19 and related vaccination strategies, added quantitative results and outputs where needed, remade the figures, and listed up-to-date references. Finally, you have polished the manuscript’s language.

Regarding the content of your manuscript, I have no more comments; regarding the form, make sure you follow all journal guidelines recommended to authors.

Kind regards,

Your reviewer